

# Socio-economic factors and management regimes as drivers of tree cover change in Nepal

Sujata Shrestha[1,2], Uttam B. Shrestha[2] and Kamal Bawa[1,3]

[1] Department of Biology, University of Massachusetts Boston, Boston, MA, United States of America
[2] Institute for Agriculture and the Environment, University of Southern Queensland, Toowoomba, QLD, Australia
[3] Ashoka Trust for Research in Ecology and the Environment (ATREE), Bangalore, India

## ABSTRACT

Despite the local and global importance of forests, deforestation is driven by various socio-economic and biophysical factors continues in many countries. In Nepal, in response to massive deforestation, the community forestry program has been implemented to reduce deforestation and support livelihoods. After four decades of its inception, the effectiveness of this program on forest cover change remains mostly unknown. This study analyses the spatial and temporal patterns of tree cover change along with a few socio-economic drivers of tree cover change to examine the effectiveness of the community forestry program for conserving forests or in reducing deforestation. We also investigate the socio-economic factors and policy responses as manifested through the community forestry program responsible for the tree cover change at the district level. The total tree cover area in the year 2000 in Nepal was ∼4,746,000 hectares, and our analysis reveals that between 2001 and 2016, Nepal has lost ∼46,000 ha and gained ∼12,200 ha of areas covered by trees with a substantial spatial and temporal variations. After accounting socio-economic drivers of forest cover change, our analysis showed that districts with the larger number of community forests had a minimum loss in tree cover, while districts with the higher proportion of vegetation covered by community forests had a maximum gain in tree cover. This indicates a positive contribution of the community forestry program to reducing deforestation and increasing tree cover.

# INTRODUCTION

Forests play multiple roles in climate regulation, protection from extreme events, water filtration, carbon sequestration, and biodiversity habitat apart from providing provisioning ecosystem services such as food, timber, and medicines (*Lambrechts, Wilkie & Rucevska, 2009*; *Anderegg, Kane & Anderegg, 2013*). Forests regulate regional and global climate through evapotranspiration, which in turn affects the precipitation regime and the water cycle (*Chagnon & Bras, 2005*). About 45% of carbon found in terrestrial ecosystems is stored in forests, and forests sequester more than 25% of annual anthropogenic carbon

Corresponding author
Uttam B. Shrestha,
Uttam.Shrestha@usq.edu.au,
ubshrestha@yahoo.com

emissions from the atmosphere (*Pan et al., 2011*). Forests, with the majority of the world's terrestrial species of plants, animals, and microorganisms, are also one of the richest biological areas on Earth (*Lindenmayer, Margules & Botkin, 2000*). Furthermore, about 1.3 billion people, primarily in developing countries, rely on forests for their subsistence livelihoods and a significant part of cash income (*Wasiq & Ahmad, 2004*).

Despite the important and critical role forests' play in maintaining essential functions of our planet and in human welfare, the process of converting forested land to other land uses such as cropland, pasture, mining, and urban areas is persistent (*Keenan et al., 2015*). Although the rate of deforestation has slowed down in recent years, it is still alarmingly high (*FAO, 2010*). About 13 million hectares (ha) of forests were lost annually from 2010 to 2015 at the global scale, and the extent of forest loss is higher in tropical countries (*Hansen et al., 2013*; *FAO, 2015*), where biological diversity as well as reliance on forests for subsistence level livelihoods are the highest. Deforestation has caused degradation of quality and amount of ecosystem services around the world reducing biodiversity, undermining the flood retention capacity and soil stability as well as producing negative impacts on local livelihoods and regional economies (*Wagner, Yap & Yap, 2015*). The global deforestation is causing a significant amount of carbon emission (8–10% of total) contributing to global climate change and environmental degradation affecting human wellbeing (*Le Quéré et al., 2016*; *Lambrechts, Wilkie & Rucevska, 2009*). Therefore, efforts for accurate monitoring of forests at different scales have received particular attention in recent years (*FAO, 2015*; *Shimada et al., 2014*).

## Forest cover change and its drivers

Deforestation is influenced by a wide range of factors such as agricultural expansion, insecure land tenure, international markets, colonization, infrastructure and road building, urbanization, mining, grazing, uncontrolled fire, political unrest, fuelwood extraction, and timber logging (*Angelsen & Kaimowitz, 1999*; *Geist & Lambin, 2002*; *Rudel et al., 2009*; *Ferretti-Gallon et al., 2014*). Various demographic, socioeconomic, biophysical, political, cultural, and technological drivers, acting individually or synergistically, stimulate the anthropogenic activities of the agents (i.e., small farmers, ranchers, plantations, loggers) causing deforestation or forest degradation (*Angelsen & Kaimowitz, 1999*; *Kissinger, Herold & Sy, 2012*). For example, an increase in human population requires more land for food, space, and other commodities resulting in the conversion of forest areas into agriculture or other land uses (*Kanninen et al., 2007*). A synthesis of more than 140 economic models analyzing the causes of tropical deforestation showed that more roads, higher agricultural prices, lower wages and a shortage of off-farm employment lead to more deforestation, while the effect of technical change, agricultural input prices, household income levels and tenure security on deforestation is unknown, and the role of macroeconomic factors such as population growth, poverty reduction, national income, economic growth, and foreign debt on deforestation is ambiguous (*Angelsen & Kaimowitz, 1999*). However, the drivers of deforestation vary across geographical locations and historical contexts; over the last 50 years, the agents of deforestation have changed (*Laurance & Balmford, 2013*). Historically, forests were cleared for crops or livestock, and small farmers were considered as

a major driver of deforestation. Conversely, after economic globalization since 1990, forests have been cleared for massive agricultural expansion, road building, wood extraction, and infrastructure development (*Rudel et al., 2009*; *Laurance & Balmford, 2013*). A large-scale agriculture expansion for cattle ranching, soybean and palm oil production, and timber logging is causing deforestation in many countries such as Brazil and Indonesia (*Brown et al., 2005*; *Morton et al., 2006*; *Arima et al., 2011*).

Various approaches were adopted globally such as the establishment of protected areas, forest restoration, protection and afforestation activities, and provision of economic incentives to reduce and prevent deforestation or forest degradation (*Brooks, Waylen & Mulder, 2012*; *Le Saout et al., 2013*). Around 15.4% of the world's land area (*Deguignet et al., 2014*) and about 24% of the total area of Nepal are covered by protected areas (*GoN, 2014b*), which have contributed to reducing deforestation and conserving forests. As an alternative to strict protection, as practiced in protected areas, community-based conservation initiatives such as community forestry programs adopted in Nepal and other developing countries also play a successful role in forest protection (*Brooks, Waylen & Mulder, 2012*; *Porter-Bolland et al., 2012*). More recently, reforestation has been a global phenomenon, and many developing countries have gone through a forest transition–a shift from net loss to a net increase of forest cover (*Meyfroidt & Lambin, 2011*). During 1990–2015, a net loss of the forest area has been slowing down and afforestation has increased at a global scale primarily in 13 tropical countries, forest transition has undergone since 1990 (*Sloan & Sayer, 2015*). Forest transitions result from various trends such as natural regeneration of forests, forest plantation, and adoption of agroforestry (*Meyfroidt & Lambin, 2011*). Migration of farmers from rural areas to urban centres and an economic shift from agriculture to industry and services stimulate forest recovery and gain (*Aide & Grau, 2004*). In the context of South Asia including Nepal, reforestation and regrowth of forest are attributed to human drivers, particularly to the devolution of forest management to local communities in the form of community forestry (*Nagendra, 2009*).

Globally, community forest management (CFM) has been considered a promising approach to sustainable forest management over the past few decades (*Arnold, 2001*). Although it has various definitions and interpretations, CFM is a government-approved form of forest management in which the rights, responsibilities, and authority for forest management rest, at least in part, with local communities (*Newton et al., 2015*). The primary aim of CFM is maintaining ecological sustainability (reduce deforestation, preserve biodiversity) while improving livelihoods of the local community (*Bowler et al., 2012*). Despite some examples of CFM failures (*Tole, 2010*; *Bowler et al., 2012*), in many countries, it has produced successful outcomes such as improvement of forest cover, increase in plantation zones, equity of benefit sharing, or reduction of community poverty (*Pagdee, Kim & Daugherty, 2006*). In some tropical countries, the community managed forest plays more important role in maintaining forest cover than protected areas; community forestry has lower deforestation rates than protected areas do (*Porter-Bolland et al., 2012*). Despite these mixed outcomes, CFM is the widespread approach to forest management in developing countries, including Nepal. In the context of climate change, CFM is now viewed as an option to reduce greenhouse gas emission through REDD

+ (Reducing Emissions from Deforestation and Forest Degradation), a global climate change mitigation mechanism, which has been under negotiation by the United Nations Framework Convention on Climate Change (UNFCCC) (*Agrawal & Angelsen, 2009*).

Nepal has a promising history of forest management and shows an excellent example of community-based forest conservation globally although the country has only 5.96 million hectares forests (40.36% of the country's land area). Concerned with massive deforestation and forest degradation in the early 1970s, Nepal initiated one of the most extensive community forestry (CF) programs in the world by handing government-controlled forests over to community forestry users groups (CFUGs) formed by local communities through an enactment of the Panchayat Forest Rules (*Acharya, 2002*). Since then, about 1.8 million ha of the forest areas have been handed over to and managed by, 19,361 CFUGs (approximately 1.45 million households or 35% of Nepal's population) under community-based forest management program (*DoF, 2015*). The community forests provide various ecosystem goods and services to the local communities and help to global communities by sequestering a significant amount of carbon. Nepal has recently joined to the United Nations collaborative initiative on REDD + program–one of the leading global efforts to reduce deforestation and mitigate climate change, prepared a Readiness Preparation Proposal (R-PP), and formed REDD+ institutional framework (*MoFSC, 2012*; *UN-REDD/REDD Cell, 2014*). Finally, under the most recent United Nations Framework Convention on Climate Change (UNFCCC) agreement in Paris to reduce emissions, Nepal's Intended Nationally Determined Contributions (INDCs) to emissions assign a vital role to forests. Nepal aims to enhance forest carbon stock by 5% by 2025 as compared to 2015 levels and decrease mean annual deforestation rates by 0.05% from 0.44% (*DFRS, 2015*).

Nepal does not have a long-recorded history of deforestation or forest cover change. The initiation of large-scale monitoring of forest cover change occurred only after 1960 although deforestation has been a major issue in Nepal. From 1964 to 1994, about 2.1 million ha of forests were converted to shrubland or other land uses (*Acharya et al., 2015*). FAO data showed that the annual loss of forest in the period between 2000 and 2005 was 1.39%, which remained stable during 2005–2010 (*FAO, 2010*). However, forest cover change at the national level has not been assessed in Nepal after the second National Forest Inventory in 1999; therefore, there is no critical information available about forest cover change at the national level in Nepal in recent years (*DFRS, 2015*). Most of the recent studies on forest cover change were conducted in small areas (*Uddin et al., 2015a*; *Uddin et al., 2015b*; *Niraula et al., 2013*; *Poudel, Fuwa & Otsuka, 2015*). Some recent studies have outlined both drivers and underlying causes of deforestation and forest degradation in Nepal. A total of nine major drivers of deforestation and forest degradation: (i) forest fire, (ii) overgrazing, (iii) unsustainable utilization of forest products, (iv) weak forest management practices, (v) infrastructure development, (vi) urbanization and resettlement, (vii) encroachment, (vii) invasive species and (ix) mining were identified (*REDD Implementation Center, 2013*). Likewise, population distribution, migration, poverty, high dependency in forest products, insecure forest tenure are major underlying causes of deforestation and forest degradation in Nepal (*Acharya et al., 2015*). We use available data on the drivers and underlying causes of deforestation and analyse: (a) the spatial and temporal patterns of the tree cover change

in Nepal from 2001–2016, (b) the socio-economic drivers of forest cover change, and (c) the effectiveness of community forestry programs on the tree cover dynamics. Our efforts mark the first attempt to analyse the tree cover change for the entire country (albeit at the district level), relate this loss and gain to socio-economic drivers, and identify policy-relevant interventions (community forestry) needed to stem deforestation and forest conservation.

## MATERIALS AND METHODS

### Study area

Nepal provides an excellent case study to understand the effectiveness of community-based institutions on forest conservation as a significant portion of the forests of this country is managed by local communities. The entire country, Nepal (Fig. 1) with a geographical area of 147,181 km$^2$, was divided into five physiographic zones–High Himal, High Mountain, Hill, Siwalik, and Terai–based on climate, soil, elevation, topography, vegetation and forest types (LRMP, 1986). Forest and agriculture sectors have the highest contributions to the national gross domestic product (GDP), contributing 26.1% share of the total GDP of Nepal (CBS, 2015b). About 69% of the employed population in Nepal is engaged in agriculture, forestry, and fishing (CBS, 2015a). The country is politically divided into 75 districts; the district is the lowest spatial unit used here for data analysis, as most of the demographic, socio-economic, and environmental data are available only at the district level in Nepal. We believe that district-level analysis does not compromise data availability and spatial accuracy.

### Tree cover data

We used a subset of global tree cover data provided by global forest watch (Hansen et al., 2013, updated every year). The global forest watch offers the highest resolution datasets (30 m ground resolution) of tree cover using Google Earth Engine and Landsat's satellite imagery for the entire globe. The data show both the extent and change of tree cover globally (Hansen et al., 2013). We called it tree cover; however, it is synonymously called forest cover. Unfortunately, there is no shared definition of forests globally. Generally, forests are defined for specific purposes, based on views, concepts, and priorities (Chazdon et al., 2016). Three common criteria: canopy cover, intact-area, and the height of the trees are commonly used for defining forests, but these criteria are not uniformly used by different agencies and countries. For example, Food and Agriculture Organization of the United Nations (FAO) uses 5 m for the height of trees, 10% crown cover and 0.5 ha for minimum size of forest (Lambrechts, Wilkie & Rucevska, 2009) whereas UNFCCC calls forests for area of 0.05–1 ha with 10–30 % canopy and >2–5 m tall trees (Sasaki & Putz, 2009). These differences in definitions and methodologies used to map and monitor forests often lead to differing results (Lambrechts, Wilkie & Rucevska, 2009). Furthermore, the definition and assessment issues have handicapped the efforts to understand the tree cover dynamics (Rudel et al., 2016). The definition of tree cover adopted for this study was 'all the vegetation area greater than 5 m in height with the canopy cover of at least 30%' as used by Hansen et al. (2013) since we used Hansen's data from the Global Forest

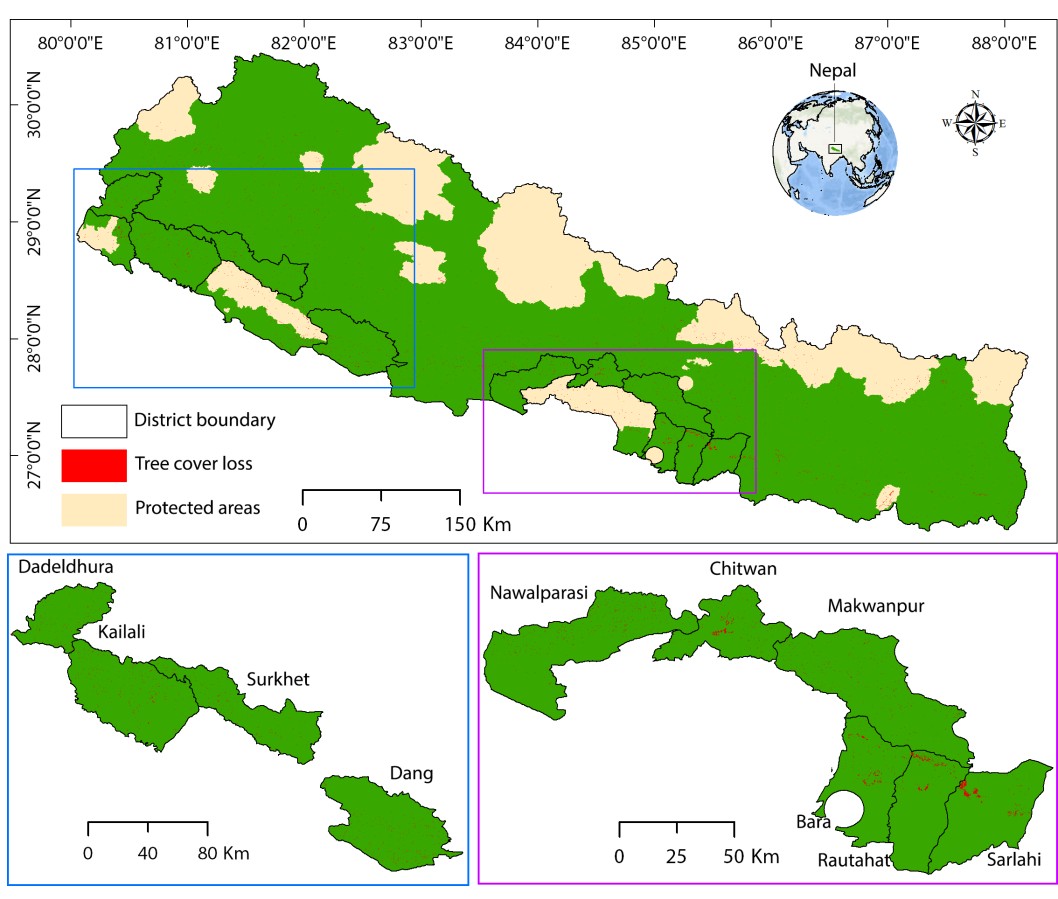

**Figure 1** Study area showing the tree cover loss in different districts of Nepal.

Watch. We disaggregated tree cover change (loss and gain) data for the 75 districts and five physiographic zones of Nepal.

## Drivers of tree cover change and policy responses

According to *Angelsen & Kaimowitz (1999)*, the framework of deforestation should consider five types of variables (the magnitude and location of deforestation as a dependent variable, the agents of deforestation, the choice variables, agents' decision parameters, the macroeconomic variables, and policy instruments) in the models of deforestation. We selected a list of potential factors of deforestation and forest degradation in the context of Nepal after carefully reviewing the literature on global and local drivers of–and causes to–forest cover change, while also considering the availability of data. We did not consider some factors associated with deforestation and forest degradation identified by previous studies (*Angelsen & Kaimowitz, 1999*) as relevant to Nepal. For example, some of the immediate causes of deforestation listed by *Angelsen & Kaimowitz (1999)* such as agriculture prices, prices of agricultural inputs and credit and underlying causes of deforestation such as timber prices, external debt, trade and structural adjustment may be associated with deforestation in Nepal but the data is not available.

Data on various factors associated with deforestation and forest degradation such as demographic (population density, migration), economic (poor people density, livestock density), social (human development index, percentage of households using fuelwood for cooking), and environmental (fire, road length, elevation, slope) along with the policy response variables (number of community forest user groups and the share of the major vegetation area covered by community forest) were gathered from various sources (Table 1). In the following paragraphs, we provide our justification for selecting these factors.

The population is widely seen as an underlying driver of deforestation (*Angelsen & Kaimowitz, 1999*; *Kissinger, Herold & Sy, 2012*). In the natural resource-dependent country like Nepal, population growth increases demand for natural resources primarily forests and requires more lands for habitation. As the population grows, more people are living in the cities, and urbanization is considered as one of the major drivers of deforestation in Nepal (*REDD Implementation Center, 2013*). A high unemployment rate coupled with population growth has accelerated both domestic and international migrations in Nepal, and the country has emerged as a remittance-dependent economy shaped by the earnings of labour migrants for foreign employment. Remittance contributes around 29% to the Gross Domestic Product (GDP) of Nepal, making the nation top third among the countries in terms of remittance contributions to GDP (*World Bank, 2016*). Migration particularly labour migration has resulted in land abandonment and the conversion of agricultural land into other land use such as forests, shrubs or fallow in some areas (*Paudel, Tamang & Shrestha, 2014*). Therefore, population density and migration are considered as dependent variables in our model. We used the most recent national population census as a demographic variable (*CBS, 2012*). We also included the data on out-migration (the number of people migrated abroad for employment as there is no data on internal migration available) (*GoN, 2014a*). To normalize the effect of the district size, we calculated the number of migrant per unit area in a district by dividing the total number with the area of a district outside the protected areas and used in the model.

Human development index, which measures income, health, and education, is linked with deforestation and hence incorporated in our model; countries with low HDI has a high rate of deforestation and vice versa (*Jha & Bawa, 2006*). Although income and poverty are correlated, poverty is a multidimensional social phenomenon (*Anand & Sen, 1997*). There is a high rate of poverty in naturally forest-dense areas, and poverty is considered as an important underlying cause of forest conversion by smallholders (*Chakravarty et al., 2012*). However, the linkage between poverty and forest degradation is ambiguous (*Angelsen & Kaimowitz, 1999*) and the natural resource degradation may depend on a complex range of choices and tradeoffs available to the poor (*Barbier, 2010*). Nepal still has a large number of people living in poverty and has a low score in HDI. Therefore, we accounted for HDI and poverty in our model. District wise figures of HDI and number of poor people obtained from the latest human development report (*NPC/UNDP, 2014*). To normalize the effect of the district size, we calculated the number of poor people per unit area (poor people density) in a district and used in the model.

Overgrazing is considered as one of the major drivers of forest loss and degradation in Nepal (*Acharya et al., 2015*; *REDD Implementation Center, 2013*). Grazing in the forested

**Table 1  Description of the variables used in the regression model.**

| Variables | Description | Unit | Data source |
|---|---|---|---|
| Ncfug | Number of community forestry user groups (CFUGs). CFUG is a community based local institution that has right to manage and govern community forests in Nepal's community forestry program. | Number | Community Forestry National Database Department of Forest, Government of Nepal (*DoF, 2015*) http://dof.gov.np/image/data/Community%20Forestry/Detail%20FUG%20All.pdf |
| Rcfug | Percentage of major vegetation area (cumulative of trees, grasslands, shrubs and sparse vegetation areas potential to be community forests) in the district covered by community forests. | Percentage | Calculated based on the land cover map and area of community forest in the district |
| Pdensity | Population density in 2011 (calculated by dividing population with the area of a district outside the protected areas) | Number/ha | Central Bureau of Statistics (CBS), Government of Nepal (*CBS, 2012*) http://cbs.gov.np/image/data/Population/District%20Level%20Detail%20Report/Household_Tables.pdf |
| Ppopdensity | Density of poor people (calculated by dividing population poor people with the area of a district outside the protected areas) | Number/ha | Calculated here |
| Rlivest | Ratio of total number of livestock with the extent of major vegetation area in a district | Number/ha | Calculated here from the data gathered from Promotion and Statistics Division, Ministry of Agricultural Development, Government of Nepal (*GoN, 2012a*) http://www.moad.gov.np/en/publication?PublicationSearch%5Bcategory_id%5D=13&PublicationSearch%5Btitle%5D=&PublicationSearch%5Badded_date%5D= |
| Hdi | Human development index (composite index of life expectancy, education and per capita income) | | National Planning Commission, Government of Nepal (*Newton et al., 2015*) http://www.hdr.undp.org/sites/default/files/nepal_nhdr_2014-final.pdf |
| Rlength | Total length of roads | Km | Department of Roads, Government of Nepal (*GoN, 2012b*) http://dor.gov.np/home/page/road-statics-2013-14-1 |
| Fire | Total number of fire (pixel) incidence (clipped by the extent of vegetation) from 2001 to 2016 divided by the area of a district outside the protected areas | Number/ha | http://modis-fire.umd.edu/pages/BurnedArea.php?target=GeoTIFF |
| Migrant | Number of migrants from the district gone to overseas for employment (2008–2014) | Number | Department of Foreign Employment, Government of Nepal (*GoN, 2014a*; *GoN, 2014b*) https://asiafoundation.org/resources/pdfs/MigrationReportbyGovernmentofNepal.pdf |
| Fuelwood | Percentage of households using fuelwood for cooking in the district | Percentage | Calculate here from the data gathered from National Population and Housing Census (National Report), Central Bureau of Statistics (CBS), Government of Nepal (*CBS, 2012*) http://cbs.gov.np/sectoral_statistics/population/national_report |
| Tloss | Net change loss in tree cover from 2001–2016 | Hectare (ha) | Global Forest Watch (*Hansen et al., 2013*) http://earthenginepartners.appspot.com/science-2013-global-forest/download_v1.4.html |

.

| Variables | Description | Unit | Data source |
|-----------|-------------|------|-------------|
| Tgain | Net gain in tree cover from 2001–2016 | Hectare (ha) | Global Forest Watch (*Hansen et al., 2013*) http://earthenginepartners.appspot.com/science-2013-global-forest/download_v1.4.html |
| Tchange | Net change in tree cover from 2001–2016 | Hectare (ha) | Calculated here (net gain-net loss) |
| Elevation | Altitude | Meter | Shuttle Radar Topographic Mission (SRTM) Digital Elevation Data (DEMs) https://lta.cr.usgs.gov/SRTM1Arc |
| Slope | Slope | Degree | Calculated from elevation in ArcGIS |

areas and stripping trees to provide fodder for animals are common in many parts of Nepal. Therefore, we considered this as a variable to our model. The most recent data of livestock were acquired from the statistical information on Nepalese Agriculture (*GoN, 2012a*). Since the populations of pig, poultry, and fowl do not have a direct impact through grazing on forests, we excluded them from the populations of cattle, buffalo, sheep, and goat and used as the livestock number. We calculated the livestock ratio by dividing the livestock number with the extent of the major vegetation cover of each district assuming the pressure of these livestock exerts mainly on vegetation. The vegetation area (cumulative area of forests, shrubs, grasslands and sparse vegetation) for a district was calculated by using a global land cover share map, version 1.0 (*Latham et al., 2014*).

Although fire can be a helpful tool for forest management, it can be a significant cause of deforestation if abused (*Chakravarty et al., 2012*). Forest fire induced by humans is one of the key drivers of forest degradation in Nepal (*Matin et al., 2017*). We used the burned area products (MCD45) of the MODIS satellites. We calculated a yearly total number of fire pixels from the monthly images of Geotiff version. Annual fire incidence (number of pixels) was then clipped by the extent of vegetation to exclude fire occurred outside the potential forest area. Total pixels of the yearly fire incidences were counted in each district. We later combined all the annual values from 2001–2016 and the aggregated value was used in the model.

Fuelwood gathering is considered as one of the causes of deforestation and forest degradation in tropical areas (*Chakravarty et al., 2012*). In rural areas of Nepal, wood derived from natural forests is one of the most critical sources of fuelwood (*Christensen, Rayamajhi & Meilby, 2009*). Fuelwood contributes about 70% of the total energy supply for the rural population of Nepal (*Kandel et al., 2016*). Therefore, we included fuelwood gathering as a variable in our model. We collected district wise data of the total number of households used fuelwood for cooking from the national population and housing census (*CBS, 2012*). We calculated the percentage of households in a district using fuelwood for cooking to use as a predictor in the model.

Proximity to the roads affects forest condition; forests closer to roads in the distance are more likely to be cleared (*Liu, Iverson & Brown, 1993*; *Lambin, 1997*). In rural Nepal, there has been a prolific growth of earthen road expansion in recent years. Due to the mountainous topography, steep slopes, and weak soils, these poorly constructed rural roads have increased the probability of landslides especially during the heavy monsoonal

rainfall (*Leibundgut et al., 2016*). Therefore, road buildings may have an impact on the condition of forests and we considered the length of the road as a variable in our model. The data on roads were collected from the Department of Roads, Nepal (*GoN, 2012b*). We also used digital elevation data (DEM) from Shuttle Radar Topographic Mission (SRTM) (http://srtm.csi.cgiar.org/) and calculated slope from DEM in ArcGIS to use in our regression model.

We used the total number of CFUGs and the proportion of the vegetation area covered by community forests in a district as a proxy to measure the effectiveness of community forestry programs. The data on the number and area of CFUGs were obtained from the Management Information System maintained by the Department of Forests, Nepal (*DoF, 2015*). To normalize the non-forested area effect, the total area of community forests in a district was divided by the major vegetation area of that district because the government handed over only the area covered by potential vegetation (forests, grasslands, shrubs, and sparse vegetation areas) to the local community as the community forests.

## Data analysis

We analysed the net change, loss, and gain of tree cover for each district over a 16-year period from 2001 to 2016. Because protected areas cover a significant area of Nepal (about 24% of the total land area) and have a separate management system, the geographical areas covered by protected areas were excluded in further analysis to determine the impacts of the drivers of deforestation and effectiveness of the community forestry program on the gain and loss of tree cover. We build three models: in the first model, the proportion of tree cover loss was used as a dependent variable; in the second, proportion of tree cover gain; and the proportion of net change in tree cover in the third model. The demographic, economic, social, and environmental variables were used as independent variables in all three models. After testing our data with the assumptions required for multiple linear regressions (heteroscedasticity, normality, outliers, multicollinearity), we conducted the ordinary least square (OLS) regression analysis to predict the impact of independent variables on the dependent variables. We examined multicollinearity among predictor variables (Fig. S1) and eliminated highly correlated ($r > 0.75$) two variables (poor people density and slope) resulting in 10 independent variables for the initial models. We used stepwise model selection method on R software package to select the final model (*R Core Team, 2017*). The initial models were evaluated by using Akaike Information Criteria (AIC), the commonly applied criterion to compare models for the goodness of fit and the model with the smallest AIC was chosen as the best-fit model (*Burnham & Anderson, 2004*).

As the socio-economic data were available at the district level, we choose 75 districts as study units. We conducted area-based correction for the dependent variables (tree cover loss and gain) to normalize the effect of the district size. Rather than using total area of tree cover loss and gain, we used proportions of forest that were lost, gained and changed in the district as the dependent variable in the regression models.

We also quantified the spatial pattern of tree cover loss to observe the spatial association between roads and the loss and gain of tree cover, using the GIS-based buffering approach, from one to five-kilometre distance from the current road networks. We counted the total
number of pixels of tree cover loss and gain within a range from one to five kilometres from the roads and calculated the total areas. Since we have temporal data of forest fire incidence and forest fire is considered a major driver of deforestation and forest degradation in Nepal, we also compared annual trends of tree cover loss with the trends of the forest fire.

We visually compared the tree cover loss data with the high-resolution images of Google Earth Pro (https://www.google.com/earth/download/gep/agree.html). We first identified 132 larger patches of tree cover loss and randomly selected 50 patches by overlaying 1 km$^2$ grids on the layer of tree cover loss. We visually compared the images captured around 2001 with the images captured around 2016 in those loss patches using Google Earth Pro. About 71.7% time, the tree cover loss patches matched with actual loss of tree cover (Fig. S2).

## RESULTS

### Spatial pattern of forest cover change

The total tree cover area in the year 2000 in Nepal was 4,746,000 hectares. Nepal has lost 46,000 ha and has gained 12,200 ha areas of tree cover over the last 16 years from 2001–2016. However, a substantial spatial variation was observed among physiographic zones, and districts; maximum loss of tree cover in Siwalik (28%, 13,000 ha) followed by Hill (26%, 12,100 ha) and Terai (22%, 9,900 ha), Middle mountain (21%, 98,00 ha) and High mountain (2%, 1,100 ha). Regarding tree cover gain, the Hill region gained the highest area of 6,200 ha (51%) followed by Siwalik 3,000 ha (25%), Terai 2,100 ha (17%), Middle mountain 830 ha (7%), and High mountain 70 ha (1%).

A major loss in tree cover was observed in Kailali (6%, 2,270 ha), Dang (5%, 2,090 ha), Sarlahi (4%, 1,730 ha), Rautahat (3%, 1,260 ha) and Nawalparasi (3%, 1,180 ha) districts whereas Kaski (0.09%, 34 ha) and Bhaktapur (0.1%, 42 ha) lost comparatively a smaller area of tree (Fig. 2). In terms of gain in tree cover, Dang with 880 ha (7%) forest gain was at the top position followed by Nawalparasi (7%, 830 ha), Tanahun (6%, 650 ha), Palpa (6%, 620 ha) and Kailali (5%, 600 ha) districts while Manang (0.01%, 1 ha), Kaski (0.01%, 1 ha) and Darchula (0.01%, 1 ha) gained a lesser forest area. The maximum loss and gain of tree cover were observed within the five-kilometre distance from the roads; the area of forest cover loss and gain decreased as the distance from the roads increased (Fig. 3A).

### Temporal pattern of forest cover change

Over 2001–2016, the maximum loss in tree cover (6,180 ha) occurred in the year 2009 and the minimum (1,040 ha) in the year 2015 (Fig. 3B). Likewise, in different physiographic regions, the maximum loss of tree cover in Terai occurred in 2009, Siwalik in 2011, Hill in 2012, Middle mountain and High mountain in 2009. There was no significant correlation between the annual incidence of the forest fire expressed as the number of pixels with the annual loss of tree cover ($r^2 = 0.037$, $p = 0.473$) (Fig. 3C).

### Drivers of change in tree cover

We observed the associations between policy response variables (proportion of the major vegetation area covered by community forest and number of CFUGs) and the proportion of tree cover loss and gain by incorporating the effects of demographic factors, economic,

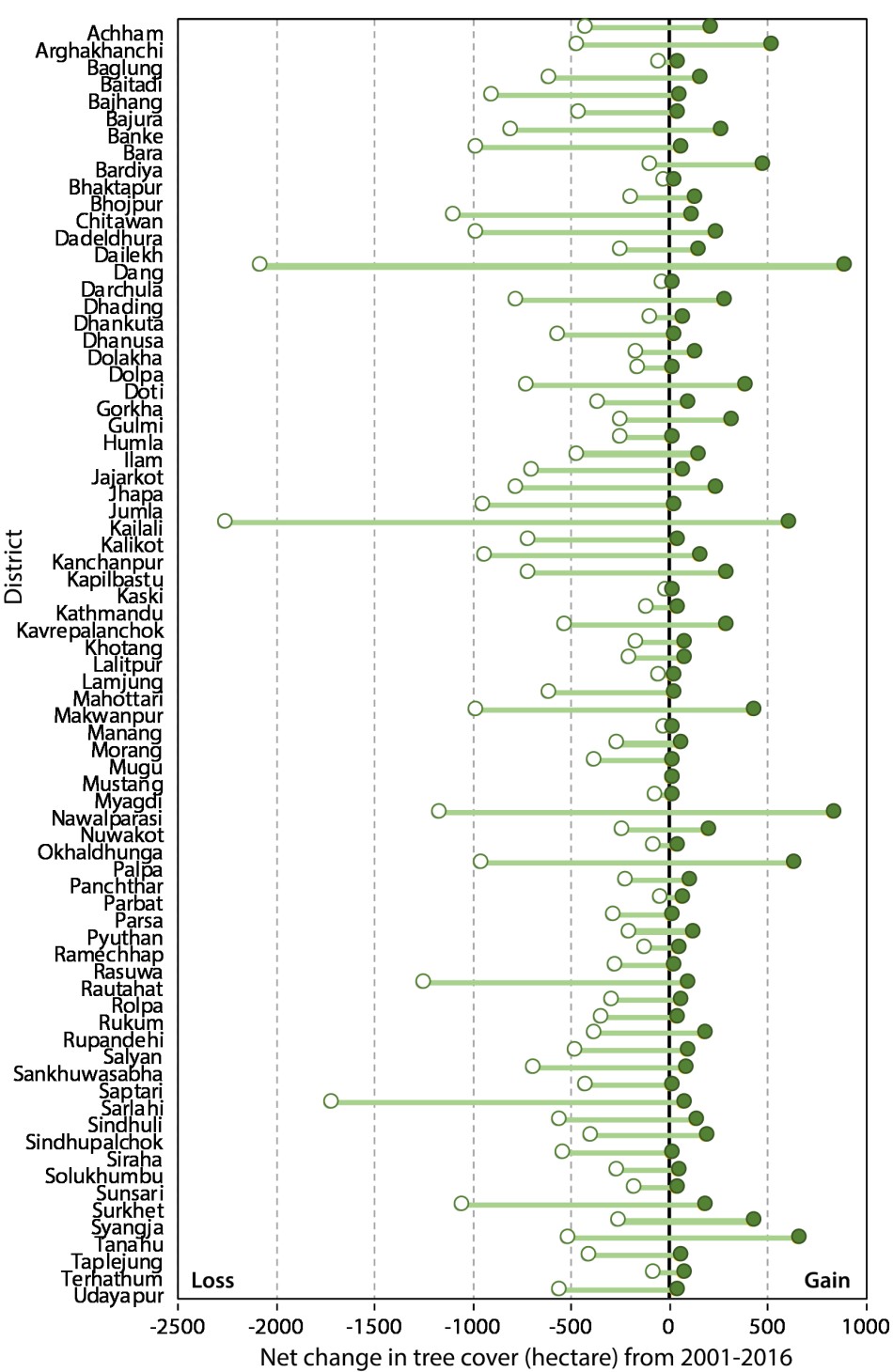

**Figure 2** Extent of tree cover change in different districts of Nepal.

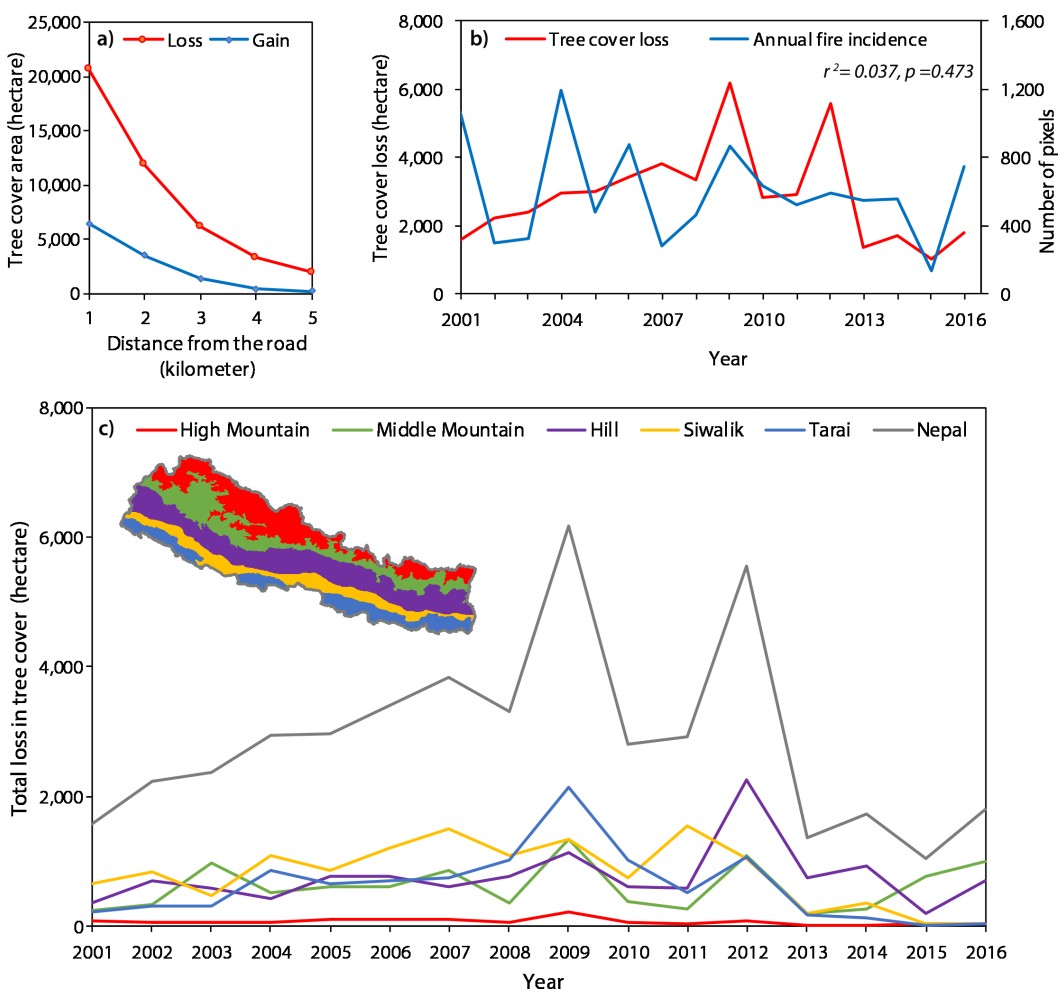

**Figure 3** (A) Tree cover loss and gain in distance from the road, (B) Temporal pattern of tree cover loss in five physiographic regions of Nepal, (C) Temporal pattern of tree cover loss with respect to forest fire incidence.

social, and environmental factors. The predictor variables identified by the AIC criterion are: number of community forests, livestock density, HDI, migrant density, and elevation for the tree cover loss model. Likewise, a proportion of area covered by community forests, elevation and road lengh were retained for tree cover gain model (Table 2). In the net change in tree cover model, a proportion of area covered by community forests, number of community forests, livestock density, HDI, migrant density, and elevation were retained out of ten predictors. According to our model, the proportion of tree cover loss in a district is significant and negatively correlated with the number of community forests in the district suggesting the districts with a higher number of community forests have a lower amount of tree cover loss (Table 2). Similarly, HDI and elevation had a significantly negative correlation with tree cover loss while migrant density had a positive association.

Similarly, the area of tree cover gain was significant and positively associated with the proportion of the major vegetation area covered by community forests demonstrating

**Table 2  Regression models for the forest cover loss, gain and net change.**

| Model 1 Proportion of forest cover loss | | Model 2 Proportion of forest cover gain | | Model 3 Proportion of net forest cover change | |
|---|---|---|---|---|---|
| Predictors | Estimate (Std. Error) | Predictors | Estimate (Std. Error) | Predictors | Estimate (Std. Error) |
| (Intercept) | 4.76900 (1.21400)*** | (Intercept) | 0.06492 (0.16150) | (Intercept) | 5.59100 (1.2)*** |
| Ncfug | −0.00350 (0.00090)*** | | | Ncfug | −0.00390 (0.00090)*** |
| | | Rcfug | 0.01273 (0.00309)*** | Rcfug | 0.01279 (0.00754) |
| Rlivest | −0.00007 (0.00004) | | | Rlivest | −0.00005 (0.00004) |
| Hdi | −5.75200 (2.78700)* | | | Hdi | −8.58600 (2.839)** |
| Migrant | 3.37400 (1.06000)** | | | Migrant | 2.67700 (1.047)* |
| Elevation | −0.00028 (0.00013)* | Elevation | −0.00012 (0.00005) | Elevation | −0.00021 (0.00013) |
| | | Rlength | 0.00008 (0.00005) | | |
| $R^2 = 0.4755$, $p = 0.00002$ | | $R^2 = 0.2842$, $p < 0.00001$ | | $R^2 = 0.4872$, $p < 0.00001$ | |

Notes.
*$p \leq 0.05$.
**$p \leq 0.01$.
***$p \leq 0.001$.

that districts with a higher the proportion of community forests have a more significant area of tree cover gain. In the net forest change model, the net change in forest cover was significant and negatively associated with the proportion of the major vegetation area covered by community forests and HDI and significant and positively correlated with migrant density (Table 2). HDI, migrant density, proportion of the major vegetation area covered by community forests and the number of community forests were the critical predictors of tree cover change in Nepal.

## DISCUSSION

In this study, we disaggregated tree cover (both extent and change) into five physiographic zones and 75 districts of Nepal to compare spatial patterns of tree cover gain and loss. We also observed the temporal profile of the loss in tree cover at two scales, national and regional (physiographic zones). Furthermore, this study identified the demographic, social, economic, and environmental factors of tree cover change and measured the effectiveness of the community forestry program in changing the dynamics of tree cover. Our results are highly relevant to address the socio-economic drivers of tree-cover change as well as to witness the effectiveness of the community forestry program of Nepal.

Our results on various degree of tree cover loss and gain at district level correspond with the localized studies, which found decrease in forest cover in some areas (*Uddin et al., 2015a*; *Uddin et al., 2015b*) as well as increase in forest cover in others (*Niraula et al., 2013*; *Paudel, Tamang & Shrestha, 2014*). However, loss of tree cover is more prominent than gain in Nepal at the national scale. This study also confirms the widespread anticipation of the spatial pattern of forest cover change in various physiographic zones; higher rate of deforestation and forest degradation in Siwalik and Terai and the regeneration of forest in the Hill and Middle mountain region (*GoN, 2014b*). The Terai and Siwalik regions comprise mainly tropical Sal and Mixed Broad-Leaved forest and Hill encompasses Hill Sal

forest, Schima-Castanopsis forest, Chir Pine and Chir Pine-Broad Leaved forests whereas High Mountain region has temperate forests such as Cypress, Rhododendron, Spruce, and Oak Forests (*Barnekow Lillesøo et al., 2005*). From the commercial point of view, Terai and Siwalik regions have forests with maximum market value and are hence highly prone to commercial exploitation (*Acharya et al., 2015*). In contrast, Terai region has the lowest proportion (7%) of community forests while Hill and mountain have 75% and 16% respectively (*GoN, 2013*). The total incidence of forest-fire as a whole has a negative impact on the forest, and the relationship can be observed in the temporal pattern in which the annual incidences of forest fire correspond with the annual loss in tree cover. However, the relationship between annual forest loss and annual incidences of forest fire was not significant. Furthermore, forest-fire is also considered a primary cause of forest disturbance of Nepal (*DFRS, 2015*) and about 452,000 ha of land areas including forests were burned in Nepal from 2003 to 2012 (*FAO, 2015*).

Our results indicate that the community forestry program played a crucial role in reducing deforestation (tree cover loss) and increasing the forest area (tree cover gain) at the district level. The significant and negative association between the proportion of tree cover loss and the number of community forest shows that districts with a higher number of community forests have lesser areas of loss in tree cover. Similarly, the significant and positive association between the tree cover gain and the percentage of community-forested area in the district indicates a higher proportion of community forests in the districts has a greater gain in tree cover. Community forests combine a mixture of plantations and natural forests, and in most cases, local communities protect the community-owned forests allowing natural regeneration and growth (*GoN, 2013*). Nevertheless, the tree cover data (*Hansen et al., 2013*) used here do not distinguish between trees in plantations and natural forests. Therefore, it is not possible to differentiate between regenerating forests due to plantations or from the natural forests. Our study validates the local level studies (*Niraula et al., 2013*; *Gautam, Shivakoti & Webb, 2004*; *Gautam, Webb & Eiumnoh, 2002*) and widespread perception that community forestry has a positive impact on the forest cover change by reducing the loss and increasing the gain in forest areas at the district level. Furthermore, an analysis of the CFUGs reports based on the perception of the user groups at national level showed that 79% of the CFUGs reported an overall increase in tree density in the community forests (*GoN, 2013*).

There are other factors that are statistically associated with the tree cover loss such as migrant density; the higher density of migrant workers in a district, the greater the areas of tree cover loss. The positive association between forest loss and migrant density makes sense as the temporary migration for employment may increase demand for natural resources as the households of migrants receive remittance that may prompt construction activities creating more pressure to forests. Significant and negative association between the tree cover loss and HDI values indicates the districts with higher levels of development has lesser tree cover loss. Studies found that HDI as a crucial predictor of forest transition (*Redo et al., 2012*) and a lower rate of deforestation (*Jha & Bawa, 2006*). Lack of development and economic opportunities in the districts with low HDI may make people rely heavily on forest resources for subsistence use (*Angelsen et al., 2014*; *Belcher, Achdiawan & Dewi,*

*2015*) that might lead to the extraction of more forest resources causing deforestation and forest degradation.

Although we didn't find any correlation between road length and the change in tree cover, spatial concomitance between road and forest grain and loss might lead to a positive relationship between forest cover loss and road network. Construction of the road might lead to the cutting of trees and facilitate forest encroachment. Due to the steepness in the hilly areas, construction of unplanned earthen roads triggers landslides causing loss of vegetation (*Leibundgut et al., 2016*). Furthermore, roads were recently built in many rural villages of Nepal, and most of the community forests are located near villages in Nepal. Although fire incidence was considered as a driver of deforestation in Nepal, we didn't find any association between fire incidence and tree cover change. The temporal trend of total annual incidence of forest fire and forest cover loss was very weak. More detail study is necessary to understand the impacts of fire on forests in Nepal, as in this study the unit of analysis is district and there could be a mismatch between the precise localities of forest fire and the tree cover locally.

The significantly negative correlation between elevation and forest loss suggests the lowlands have the higher proportion of deforestation. This finding concures with the spatial pattern of the deforestation among physiographic zones; Terai and Siwalik, which are lowland areas of Nepal have the higher amount of tree cover loss. The lowland areas havor trees with high commercial values such as tropical Sal that is highly prone to logging (*GoN, 2014b*). *Bhattarai, Conway & Yousef (2009)* found that the amount of deforestation in Nepal particularly central Nepal decreased as the elevation increased and our result also confirms their finding.

## CONCLUSION

This study has compromised the spatial accuracy of the higher resolution of data with the availability of data at some extends. An analysis at a finer spatial scale would have produced a more nuanced view. Unfortunately, there is no spatial information (maps with boundary) available for all the CFUGs in Nepal. Although the analysis at the village development committees (VDCs), the lowest political unit of Nepal could provide more detail overview of tree cover change, the information on socio-economic drivers is available only at a district level. Despite this shortcoming due to limitations in data availability, our study has highlighted the different factors of deforestation and the effectiveness of the major forest conservation policy in Nepal albeit at a coarse scale. Because of these limitations, the inference of a robust causal relationship between the dependent and independent variables is rather difficult. Globally, the data of tree cover loss provided by *Hansen et al. (2013)* was correct only 75% of the time (*Weisse & Peterson, 2015*) and the data do not differentiate temporary and permanent loss of tree cover between natural forests or tree plantations (*Harris et al., 2016*). The quality of socio-economic data of developing countries is often criticized (*Meyfroidt & Lambin, 2008*). Given our limited understanding of the forest cover change in Nepal, the results of this study are useful in formulating policies and programs to address the drivers of deforestation and persistently improve

the existing policy on community forestry. We hope that future research with a higher resolution of demographic and socio-economic data (at the scale of community forest) can provide more nuanced results and may identify additional factors associated with forest cover change in Nepal.

Despite some shortcomings due to limitations in data availability and quality, this paper analyses the spatial and temporal patterns of tree cover loss and gain in the light of socio-economic drivers and effectiveness of one of the major forest conservation policies of Nepal. This study addresses a long-term standing policy question regarding the effectiveness of community forestry programs and reveals the likely socio-economic drivers of tree cover change in Nepal. Our results confirm that both the extent of community forestry and the number of CFUGs have positive impacts on the forests. Districts with a higher number of community forests have a minimum loss in tree cover, and the districts with a higher percentage of community forest area have a maximum gain in tree cover. Although the community forestry program has a positive impact on the forest cover by reducing the forest loss and increasing the gain, the other drivers of forest loss have been leading to the overall decline in forest area in Nepal. Nepal lost almost 46,000 ha forest area while Nepal gained roughly 12,200 ha over 2001–2016. Therefore, in order to conserve forest areas in Nepal, the current policy can be continued and improved if necessary, coupled with addressing the underlying cause of deforestation.

## ACKNOWLEDGEMENTS

We thank Dr. Hemant Ojha and anonymous reviewers for their valuable comments, which helped to substantially improve this manuscript. We also acknowledge Global Forest Watch for providing tree cover data.

### Funding
This research is funded by Rufford Small Grants for Nature Conservation, Nancy Goranson Endowment Fund, and Doctoral Dissertation Research Grant from the University of Massachusetts Boston. There was no additional external funding received for this study. The funders had no role in study design, data collection and analysis, decision to publish, or preparation of the manuscript.

### Grant Disclosures
The following grant information was disclosed by the authors:
Rufford Small Grants for Nature Conservation.
Nancy Goranson Endowment Fund.
Doctoral Dissertation Research Grant from the University of Massachusetts Boston.

### Competing Interests
The authors declare there are no competing interests.

## Author Contributions

- Sujata Shrestha conceived and designed the experiments, performed the experiments, contributed reagents/materials/analysis tools, prepared figures and/or tables, authored or reviewed drafts of the paper, approved the final draft.
- Uttam B. Shrestha conceived and designed the experiments, analyzed the data, prepared figures and/or tables, authored or reviewed drafts of the paper, approved the final draft.
- Kamal Bawa authored or reviewed drafts of the paper, approved the final draft.

## Data Availability

1. Number of community forestry user groups: http://dof.gov.np/image/data/Community%20Forestry/Detail%20FUG%20All.pdf

2. Population: http://cbs.gov.np/image/data/Population/District%20Level%20Detail%20Report/Household_Tables.pdf

3. Human development index: http://www.hdr.undp.org/sites/default/files/nepal_nhdr_2014-final.pdf

4. Livestock number: http://www.moad.gov.np/en/publication?PublicationSearch%5Bcategory_id%5D=13&PublicationSearch%5Btitle%5D=&PublicationSearch%5Badded_date%5D=

5. Road length: http://dor.gov.np/home/page/road-statics-2013-14-1

6. Fire: http://modis-fire.umd.edu/pages/BurnedArea.php?target=GeoTIFF

7. Migrant numbers: https://asiafoundation.org/resources/pdfs/MigrationReportbyGovernmentofNepal.pdf

8. Tree cover change: http://earthenginepartners.appspot.com/science-2013-global-forest/download_v1.4.html

9. Altitude: https://lta.cr.usgs.gov/SRTM1Arc.

## Supplemental Information

Supplemental information for this article can be found online at http://dx.doi.org/10.7717/peerj.4855#supplemental-information.

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
