# Peer review of "Socio-economic factors and management regimes as drivers of tree cover change in Nepal"

_PeerJ, doi:10.7717/peerj.4855_

## Round 0.1 · original submission · Major Revisions

Please attend to the frequent grammatical errors throughout (Reviewer 3).
Consider the inclusion of 2015 forest canopy cover data that is now available (as suggested by Reviewer 1).
In the introduction (Or Discussion) provide a summary of modeling methods and use of forest canopy cover data (Reviewer 2). Similarly, in the Introduction or Discussion please cover the 'state-of-knowledge' on devolved community forest management (Reviewer 3).
Provide justification for why these 15 drivers are analysed and many other potential explanatory/correlatory drivers (Reviewer 2) have not been considered in this study.
Colinearity amongst predictor (driver) variables are tested but not acted upon which means there may be redundancy in the model, that can be improved by refining the drivers investigated to an even smaller dataset (Reviewer 3 and Editor).
Reviewer 3 and the Editor are concerned that the use of change in canopy cover area rather than change in canopy cover as a percentage may lead to different and perhaps more robust findings. This may be simply a terminology issue but it is hard to tell with the brevity of methods reporting. Reviewer 3 points to other instances of the methods being too brief to understand how analysis were performed and this should be addressed.
These are simply correlatory modeled relationships and cannot be interpreted as 'causal' - instances where this is implied should be reworded accordingly.

Reviewer 1 ·

Basic reporting

This article is well written. Background information and literature are well explained and covered. Figure 1 needs north arrows and scale bars. Research questions are clearly addressed. Table 2 is not self-explainable for the variables of Ncfug, Ppop, and Rlength.

Experimental design

I'm a little concerned about the use of global forest cover data for local use though many studies use this approach. It will be helpful to directly or indirectly valid partial data with locally derived forest cover data. The manuscript mentioned 'spatial accuracy'. Further explanations would be better.

Validity of the findings

The validity of the findings depends on the accuracy of the data sources, including forest cover data. Please refer to my previous comments in box 2. The statistically methods seem no problems.

Additional comments

I have one major suggestion: the 2015 data is now available and it would be better to include the 2015 data into the analysis.

Reviewer 2 ·

Basic reporting

1. Hundreds of models have been created to access the social-economic drivers (underlying forces) of deforestation on the sub-national and national scales around the world (Angelsen and Kaimowitz, 1998). A paragraph about background and introduction should be indicated with a summary of modeling methods and using data.

Experimental design

2. Angelsen and Kaimowitz reported the existence of more than 115 different variables had been used to explain deforestation. Prior studies have identified several vital underlying forces related to deforestation: forest logging, timber price, forest land accessibility, soil suitability for agriculture and so on. Besides, political aspects, like the creation of conservation territories may also impede deforestation (Muller et al., 2012, Chomitz and Gary, 1996, Rosa et al., 2013). Why are these critical drivers not considered in this study?

Validity of the findings

3. In this study, the author(s) disaggregated tree cover (both extent and change) into five physiographic zones and 75 districts. Why the author applied the regression analysis only on an aggregated level.

Reviewer 3 ·

Basic reporting

Generally well-written, although grammar requires checking for minor revision. For example, Abstract: “in response to a massive deforestation” (L14-15); “with a huge spatial and temporal variations” (L21-22); and so on throughout.

The Introduction is interesting and provides solid context and motivation for the work, which is of global interest. However, here and/or in Discussion, I miss some global context for the success or otherwise of community-based management (CBFM) schemes. What is the (global) state of knowledge on the success of devolved forest management, especially in countries most similar to Nepal?

The structure of the article is sound. There are no original data that could be shared with the reader, and the sources that are used appear to be well referenced (although I have not checked whether every dataset could indeed be assessed via the references).

Aims are clearly stated, except that “identify policy actions needed to stem deforestation and forest degradation” does not seem to be directly addressed in subsequent sections: the policy-relevant intervention being tested here is the presence or absence of CBFM, and its correlation with rates of tree cover change; but other policy action are not assessed as far as I can tell, nor is forest degradation (as opposed to clearance) assessed directly. I suggest rewording to be more specific to what is being investigated.

Experimental design

The authors construct a multivariate model investigating the response of net change in tree cover to multiple independent variables (15 in total I think - Table 1). Due to the spatially aggregated nature of some of the variables, this analysis is conducted across N=75 districts.

The authors test for collinearity (Fig S1) but have not acted on those findings, apparently including all 15 variables and performing model selection according to AIC. I suggest that redundancy in the predictor set could be reduced, by considering both collinearity and by being more targeted in terms of the relevant units for each hypothesized effect.

For example, the response variable is *area* of tree cover change, which is presumably sensitive to district size (or at least forest extent at t0). In some cases, the authors have included both the raw numbers as an independent variable, and then also another version that is scaled in some way by area: for example, “number of community forest groups” and also the “percentage of major vegetation area … covered by community forests” (the model with the lower AIC retains the former, but not the latter). Similarly, population is included in several ways: as total district population in 2001, as the density of people outside protected areas, and as the number people below the poverty threshold.

Might the result be different if the response variable was the *proportion* of forest that was lost or gained, modelled against only those predictors that are appropriately scaled by area? Following the authors stated concerns regarding the potentially confounding effects of protected areas (L217-220), population density excludes such sites from the calculation. Similarly, I would expect the same area-based correction to apply to the number of people living below the poverty threshold (this variable is returned in the lowest-AIC model, but is currently the absolute number of people, correlating positively with the absolute area of forest change).

The authors are best-placed to decide on these matters, but I urge them to consider carefully the extent to which area-effects and collinearity might impact the inferred importance of certain variables.

In a separate analysis, the authors compare tree loss and gain at different distances from the roads. The result here are interesting (Fig 3a), however the description of methods is too brief to know exactly how these calculations were done (L231-233). Similarly, very brief methods (L233-234) accompany a temporal analysis of fire incidence and tree cover loss, which are highly correlated (Fig 3bc). It would be interesting to discuss the causes of the fires – are these deliberately set by people, to clear the vegetation? If so, is the fire incidence higher or lower under CBFM? Or are people in CBFM areas more likely to put out a naturally ignited fire? Might the multivariate model therefore include interaction terms? It is difficult to tell much here from the cross-correlations (Fig S1): the number of fires is correlated positively with the number of community based groups, but this could be due to bigger districts having both more groups and more fires than smaller districts. But then fire is negatively correlated with total population and with the number of poor people. As above, correcting according to appropriate area measurements might offer some insight.

Validity of the findings

The authors should be careful when implying causality from their current analysis, given the redundancy in variables and potentially confounding area-effects. And it would be useful if effect sizes (rather than just p-values) were discussed. For example, what impact does a unit change of CBFM (e.g. an extra group or an extra % of coverage) have on the net change in forest cover, and how does this compare to the other variables in the model? Also, were there other ‘top models’ besides the one with the lowest AIC? E.g. AIC within delta 4 of the top model?

In terms of guiding policy on the conservation of forests for biodiversity, ecosystem functioning and services, the distinction between natural forest and plantation forest should be dealt with more directly, in discussion if not in the analysis. The Hansen data do not distinguish between trees in plantations and natural forests that have been conserved or are regenerating, and as noted in the Discussion (L312-313), the community forests are often a mix of both.

Additional comments

Are different forest types more or less prone to deforestation (e.g. via grazing or fire) - could this be factored into the analysis by overlaying the land cover map with the Hansen data? And/or using physiographic region as a random grouping variable in a mixed-effects model? I offer this only as a suggestion/question, not as a criticism of the current work.

---

## Round 0.2 · Minor Revisions

Dear Uttam,

I have received a re-eview from a key reviewer.

We are both impressed with the detail and diligent response to reviewer and editor comments.

Please accept this request for minor revisions and take attention to the three points below:

Please consider the alternative variables they have suggested in square brackets. Use of these in replacement for others, may reduce the susceptibility to differences in area, and as such reduce your study to similar criticism.

Please include effect sizes described in the text, in addition to p-values.

Please include three columns in Table 2 to present not only: gain and loss, but also 'net change' (as was presenting in the original version). In addition, please ensure that net change is described in the results section and the discussion.

All the best

Steve

Reviewer 3 ·

Basic reporting

The authors have made improvements to their basic reporting. In particular, I appreciate the additional introductory text on CBFM. The manuscript would still benefit from proof-reading for minor grammatical errors

Experimental design

The authors have improved their experimental design by reducing collinearity among predictors, and by analyzing area-corrected response terms. In my opinion, the results are more robust as a result, and it is encouraging to see that CBFM measures remain important. Some of the predictors, however, remain sensitive to area, so that, for example, higher fire incidence could signal either more fires per unit area, or a larger district, or both. I believe that this remains a problem / might explain some of the stranger results (e.g. fires correlating positively with forest gain; number of user groups correlating negatively with both loss and gain of forest). Please consider the following [alternatives] and justify where and why the uncorrected variables are more meaningful predictors.

- Number of community forest user groups [Mean area covered by a user group]
- Total population [I think this is redundant, given Pdensity]
- Absolute number of poor people [ditto, given Ppopdensity]
- Total number of livestock [Mean livestock per person, interacting with Pdensity]
- Total number of households using fuelwood [% of households, interacting with Pdensity]
- Number of migrants [Mean migrants per unit area]
- Total number of fire incidence [Fire incidence per km per year]. Were fire data (MODIS) clipped to the forest extent? This might increase relevance
- Total length of roads. The unit here is given as "Km/100km" - what does this mean?

I see that (maybe in response to another reviewer's comment?), the authors no longer look at net change in forest, but instead analyse only gain and loss separately. Personally I think this is a shame, though the disaggregated results are useful too. Could proportion change, relative to the baseline, be reinstated? i.e. three columns in Table 2: gain, loss, net

Validity of the findings

Thank you for the clarifications and more nuanced interpretation of results. I would still like to see effect sizes described in the text, not just p-values - I think this is important for interpretation of how much difference CBFM potential makes. Also were other models close to the 'best models' according to AIC (e.g. < delta 4)?

Additional comments

Many thanks for the clear and detailed rebuttal

---

## Round 0.3 · accepted · Accept

Dear Uttam,

Many thanks for responding to the reviewers second round of comments in a positive fashion. Glad to see that is has improved the model and the manuscript overall. I am happy to accept this manuscript for publication in PeerJ

Congratulations
Steve

#